# Highly cross-linked carbon tube aerogels with enhanced elasticity and fatigue resistance

Lei Zhuang [1], De Lu[1], Jijun Zhang[1], Pengfei Guo[1], Lei Su [1], Yuanbin Qin[1], Peng Zhang[1], Liang Xu[1], Min Niu[1], Kang Peng [1] & Hongjie Wang [1] ✉

Carbon aerogels are elastic, mechanically robust and fatigue resistant and are known for their promising applications in the fields of soft robotics, pressure sensors etc. However, these aerogels are generally fragile and/or easily deformable, which limits their applications. Here, we report a synthesis strategy for fabricating highly compressible and fatigue-resistant aerogels by assembling interconnected carbon tubes. The carbon tube aerogels demonstrate near-zero Poisson's ratio, exhibit a maximum strength over 20 MPa and a completely recoverable strain up to 99%. They show high fatigue resistance (less than 1.5% permanent degradation after 1000 cycles at 99% strain) and are thermally stable up to 2500 °C in an Ar atmosphere. Additionally, they possess tunable conductivity and electromagnetic shielding. The combined mechanical and multi-functional properties offer an attractive material for the use in harsh environments.

Carbon aerogels (CAs) demonstrating high mechanical strength, compressive resilience, and fatigue resistance are fitting harsh environments in the industries such as heat sealing, energy dissipation, pressure sensors, and aerospace vehicles[1-3]. However, achieving these properties in a material is a critical challenge due to their mutually exclusive nature[4,5], which severely restricts the CAs' applications. Constructing robust, elastic and fatigue-resistant CAs with controllable architectures, are highly promising[6,7].

Traditional CAs, made from activated or non-graphitizable carbons, have limited deformability and are fragile[8]. There have been recent advancements in constructing CAs using flexible nano-units such as one-dimensional carbon fibers and two-dimensional graphene as structural components. By assembling these nano-units into three-dimensional porous architectures, CAs with elasticity can be realized[9-11]. The porous architectures comprising large voids facilitate the deformation of elastic struts within them. Nonetheless, the facile deformation of these struts results in relatively low mechanical strength of the corresponding CAs. For instance, carbon fiber-assembled CAs can be compressed up to 90% strain ($\varepsilon$), but their tested strength may not surpass 65 kPa[12]. Besides, CAs typically exhibit permanent deformation and strength loss after cyclic compression

(e.g., -17.3% strength loss after 100 cycles at $\varepsilon = 90\%$[13]). So far, the goals of high mechanical strength and good recoverable cyclic compressibility with minimal structural degradation have not been accomplished for CAs yet.

Biological networks, such as cobweb and bacterial cellulose made of fibrous proteins, have evolved to fulfill multiple functions, including efficient storage and dissipation of energy. Their cross-linked architectures make them one of the strongest and toughest materials with deformability ever known[14]. For such materials, force tends to transfer fast along the struct and to many others through junctions during deformation. Thus, to improve the mechanical strength and elasticity of materials, cross-links between the structs need to be rationally designed[15-18]. Polyacrylamide-based hydrogels are a typical example of cross-linked architectures that possess enhanced mechanical properties. They exhibit a maximum compressive stress over 90 kPa and their residual deformation is mere 6% after 500 cycles at $\varepsilon = 70\%$[19]. Despite much effort, researchers are still facing challenges in precisely controlling the cross-links of these materials, and so achieving the desired mechanical performance.

Apart from the design of micro-architectures, the characteristics of struts play a crucial role in determining the compressive behaviors

---

[1]State Key Laboratory for Mechanical Behavior of Materials Xi'an Jiaotong University, 710049 Xi'an, China. ✉e-mail: hjwang@xjtu.edu.cn

of CAs[20,21]. Multi-wall carbon tubes are considered one of the most suitable struts for constructing robust aerogels due to their exceptional mechanical properties. In particular, carbon tubes demonstrate high flexibility and can adapt and switch when buckling, enabling them to endure local strain while preserving their structural integrity[22]. This makes them an ideal candidate for creating high-performance CAs. Although carbon tubes have an ultra-high theoretical tensile strength reaching ~100 GPa[23], their bending and buckling strength is generally limited due to weak interwall van der Waals coupling that causes easy sliding between walls[24]. This results in near-zero friction and small energy dissipation[25], but also leads to unsatisfactory mechanical strength of corresponding aerogels.

Here, we demonstrate an approach to fabricating high-performance carbon tube aerogels (CTAs) that possess a highly cross-linked architecture. The carbon tubes, which feature $sp^2$ and $sp^3$ hybridization, exhibit improved stiffness and strength. Owing to the high cross-links and robust constituents, CTAs with a low density show enhanced mechanical properties, including a maximum compressive stress of 20.9 MPa and a completely recoverable strain up to 99%. Additionally, the CTAs we prepared demonstrate high fatigue resistance, with almost negligible permanent deformation (<1.5%) and strength loss (<2%) at $\varepsilon = 99\%$ for 1000 compressive cycles. These results are among the best reported elastic materials. Moreover, they are thermally stable in an Ar atmosphere at a temperature as high as 2500 °C. As a result, we believe that they present great potential for use in a variety of practical applications, such as high-precision pressure sensors designed to operate in severe environments.

## Results

### Fabrication and structural characterization
We employed a simple yet effective approach that combines sacrificial template and chemical vapor deposition (CVD) to prepare CTAs. Typically, conventional macroscopic aerogels constructed from nanowires or nanofibers have low densities (less than 50 mg cm$^{-3}$), which results in insufficient connections between the constituents and minimal constituent content[26]. To overcome this challenge and synthesize macroscopic CTAs with strong cross-links, we utilized silicon carbide (SiC) nanowire aerogels as raw materials[27]. The initial density of these aerogels is also low, ranging from 10 to 20 mg cm$^{-3}$, as demonstrated in Supplementary Fig. 1a; however, their density can be enhanced by subjecting them to hot-pressing. The initial aerogels that possess a highly porous isotropic architecture, are assembled by nanowires in diameters ranging from 90–160 nm, as demonstrated in Supplementary Fig. 2. Upon subjecting these raw aerogels to hot-pressing conditions (1200 °C for 2 h in Ar, at a pressure of 10–15 MPa), a laminar structure with significantly higher density of nanowires (200–300 mg cm$^{-3}$) was obtained (see Supplementary Figs. 1b and 3). The wave-like morphology observed in the laminar aerogels is hypothesized to result from uneven stress distribution during hot-pressing[28]. The presence of space between the thin lamellas may facilitate deformation of the flexible constituents upon compression. To connect the nanowires, the hot-pressed aerogels were then exposed to oxidation in a furnace at 1000 °C for 24 h in air, resulting in the transformation of SiC to SiO$_2$. This transformation also aided in the subsequent removal of ceramic nanowires through acid corrosion[29]. The laminar SiO$_2$ aerogels, obtained with a sufficient number of nanowires and cross-links, served as an ideal template for the deposition. A core-shell structure was formed when coating the laminar SiO$_2$ nanowire aerogels with carbon layers. Upon removal of the SiO$_2$ cores using hydrofluoric acid, lightweight, highly cross-linked CTAs were ultimately acquired, as shown in Fig. 1a.

The CTAs, inheriting the architecture of raw laminar nanowire aerogels, exhibit a multitude of cross-links, as illustrated in Fig. 1b, c. The hollow structure of the carbon constituents is verified by transmission electron microscope (TEM) imaging and corresponding

energy dispersive spectroscopy (EDS) mapping analysis in Fig. 1d, imparting an ultralight characteristic to the CTAs (refer to Supplementary Fig. 4, where the density of CTAs was as low as 12.5 mg cm$^{-3}$). Most synthesized carbon tubes resemble seamless, rolled graphene, with in-plane atoms bonded through localized $sp^2$ hybridization. But both theoretically and experimentally, it is proven that the concentration of defects, such as stacking faults and shearing of lattice planes, as well as the distribution of local strain, can initiate low-energy transformation windows, thereby triggering the thermodynamics of $sp^2$-to-$sp^3$ phase transition[30]. In fact, the growth of single-crystal graphene is heavily influenced by the surface conditions of template[31]. Carbon atoms are energetically favorable on top of the first-layer template based on the lowest-energy principle. In the case of our SiO$_2$ nanowire templates, numerous step edges are present on their surface. Consequently, during deposition, when graphene sheets grow and subsequently encounter, the formation of defects such as stacking faults and shearing of lattice planes occurs, as depicted in Fig. 1e. The encountered regions enduring dramatic compressive strain, my lead to a reduction in C-C interatomic and interwall distance, therefore prompting the atomic rearrangements of carbon atoms from $sp^2$ to $sp^3$ and positioning them in the most favorable locations. This is corroborated by Raman and X-ray photoelectron spectroscopy (XPS) analyses which reveal the presence of both $sp^2$ and $sp^3$ in the tubes (Fig. 1f, g).

It is noteworthy that, these $sp^2$ and $sp^3$ of carbon systems correspond to very defined orbital states[32]. $Sp^2$ is the hybridization of the atomic $2s$ orbital with $2p_x$ and $2p_y$ orbitals, resulting in a system of three planar σ-bonds forming an angle of 120°. The remaining $2p_z$ orbital that is perpendicular to the plane of the σ-bonds, forms a π-bond. $Sp^3$ corresponds to the hybridization of the atomic $2s$ orbital with its three $2p_x$, $2p_y$ and $2p_z$ orbitals in equal proportion, leading to the creation of a tetrahedral σ-bond system at an angle of 109.5°[33]. The bonding states are accompanied by unoccupied anti-bonding states σ* and π*. The electron energy loss spectroscopy (EELS) mapping in Fig. 1h shows that the π* fraction varies between 0.32 to 0.45, suggesting that the distribution of π* and σ* bands is approximately uniform along the tube (typical curves of C-K spectra are given in Supplementary Fig. 5). This observation also demonstrates that the tube walls exhibit a homogeneous blend of $sp^2$ and $sp^3$ bonds.

The joints between carbon tubes were analyzed by TEM and EELS, as presented in Supplementary Fig. 6. The TEM image (Supplementary Fig. 6a–c) reveals a welded joint between the two tubes, with an overlapped regime. This structure is speculated to result from the drop height between the original SiO$_2$ nanowire templates. As illustrated in Supplementary Fig. 6d, the growth of carbon layers on the templates attributes to the formation of bonds between atoms in the same plane, while the remaining parts of the carbon layers continue to grow along the templates, resulting in a cross joint between the tubes. Upon treated by acid corrosion, through EELS spectra (Supplementary Fig. 6e), it is validated that the SiO$_2$ templates have been removed all, leaving no Si element in the tubes. The well-connected structures observed in the CTAs may be a key determinant of their mechanical properties.

### Mechanical properties
The mechanical properties of the CTAs are shown in Fig. 2a. The typical stress vs. strain curves exhibit three characteristic deformation regions: a nearly linear elastic regime for $\varepsilon \leq 5\%$; a subsequent plateau regime for $5\% < \varepsilon \leq 80\%$; and a sharply increasing stress regime for $\varepsilon \geq 80\%$. Clearly, the fully recoverable strain of our CTAs (with a density of 12.9 mg cm$^{-3}$), can be up to $\varepsilon = 99\%$ (see the inset zoom-up curve in Fig. 2a), corresponding to an ultimately compressive stress of 8.2 MPa. The density of CTAs can be further tuned to ~20–40 mg cm$^{-3}$ by adjusting the CVD time to ~140–280 min. The prepared CTAs with the highest density of 43.2 mg cm$^{-3}$, can withstand a maximum stress of

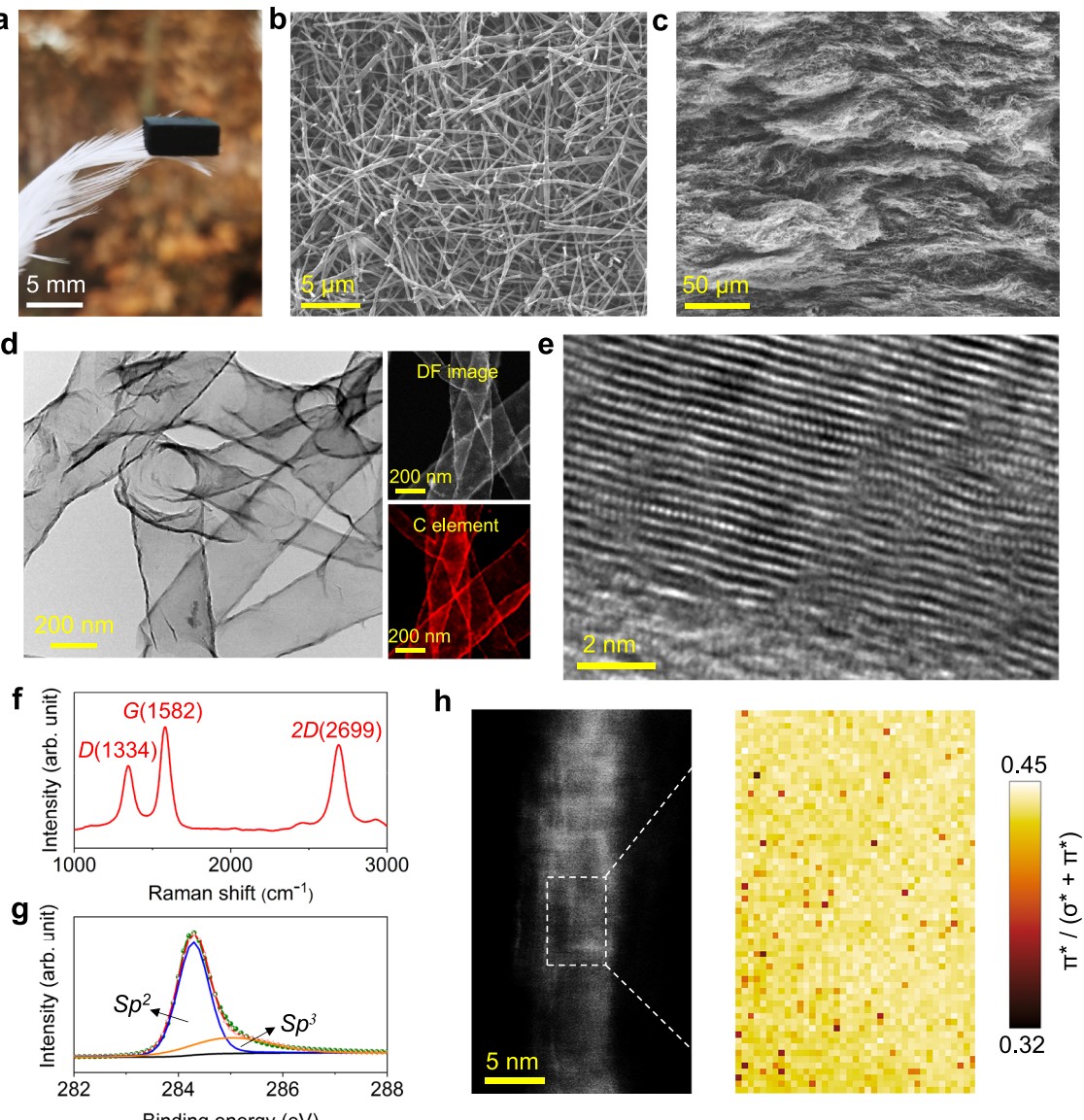

**Fig. 1 | Structure of carbon tube aerogels (CTAs). a** Lightweight characteristic of a CTA that stands on a feather. **b** Surface scanning electron microscope (SEM) image of CTAs. **c** Cross-section SEM image of CTAs. **d** Transmission electron microscope (TEM) image of the carbon tube network in CTAs. The right top and right bottom images are dark field (DF) TEM and energy dispersive spectroscopy (EDS) mapping of carbon tubes, respectively. **e** Typical high-resolution TEM of the wall of a carbon tube. **f** Raman spectrum of CTAs, showing three typical characteristic peaks of *D*, *G*, and *2D*. **g** X-ray photoelectron spectroscopy (XPS) spectrum of CTAs, showing that two types of bonds are present in the tubes. **h** Electron energy loss spectroscopy (EELS) mapping of a carbon tube wall, showing the ratio of π* to (σ* + π*) using two-window intensity-ratio method[32]. Source data are provided as a Source Data file.

20.9 MPa at $\varepsilon = 98\%$, and show mechanically recoverable behaviors with no permanent deformation as well, as presented in Supplementary Fig. 7. The architecture of highly cross-linked carbon tubes, characterized by abundant joints, enables rapid load transfer and homogenous stress distribution, leading to the good mechanical properties of CTAs.

However, excessively high cross-links (using ~300 mg cm⁻³ nanowire aerogels as template, as shown in Supplementary Fig. 8) can result in an obvious decrease in the recoverability of CTAs, despite an increase in compressive stress. This is due to the fact that the closer junctions shorten the length-to-diameter (L/D) ratio of carbon tubes, which improves the stress modulus of the tubes but also, increases their curvature when bending, ultimately reaching their maximum critical limits.

Experimental snapshots of cross-sectional views (Fig. 2b, captured from Supplementary Movie 1) and in situ SEM observations during uniaxial compression (Fig. 2c) both confirm that CTAs exhibit a near-zero Poisson's ratio. This characteristic can restrict local indirect tension, reducing permanent deformation and enhancing dimensional stability during compression.

We proceeded to assess the fatigue resistance of our CTAs. Through cyclic testing (Fig. 2d and Supplementary Fig. 9), we observe that the 100th and 1000th loops of CTAs remain nearly unchanged in comparison to the 1st cycle, displaying a maximum stress retention of up to 97.9% (8.19 MPa for the 1st cycle, and 8.02 MPa for the 1000th cycle) and a residual strain of less than 1.5% at $\varepsilon = 99\%$. The energy loss coefficient for the 1st cycle is 0.338, indicating that energy dissipation can occur by interfacial slipping between the carbon tubes during compression. After 1000 cycles, energy loss coefficient of the CTAs slightly decreases to 0.332 (Fig. 2e), demonstrating that almost no significant damage or structural collapse happens during cyclic compression. Our CTAs, possessing high mechanical strength,

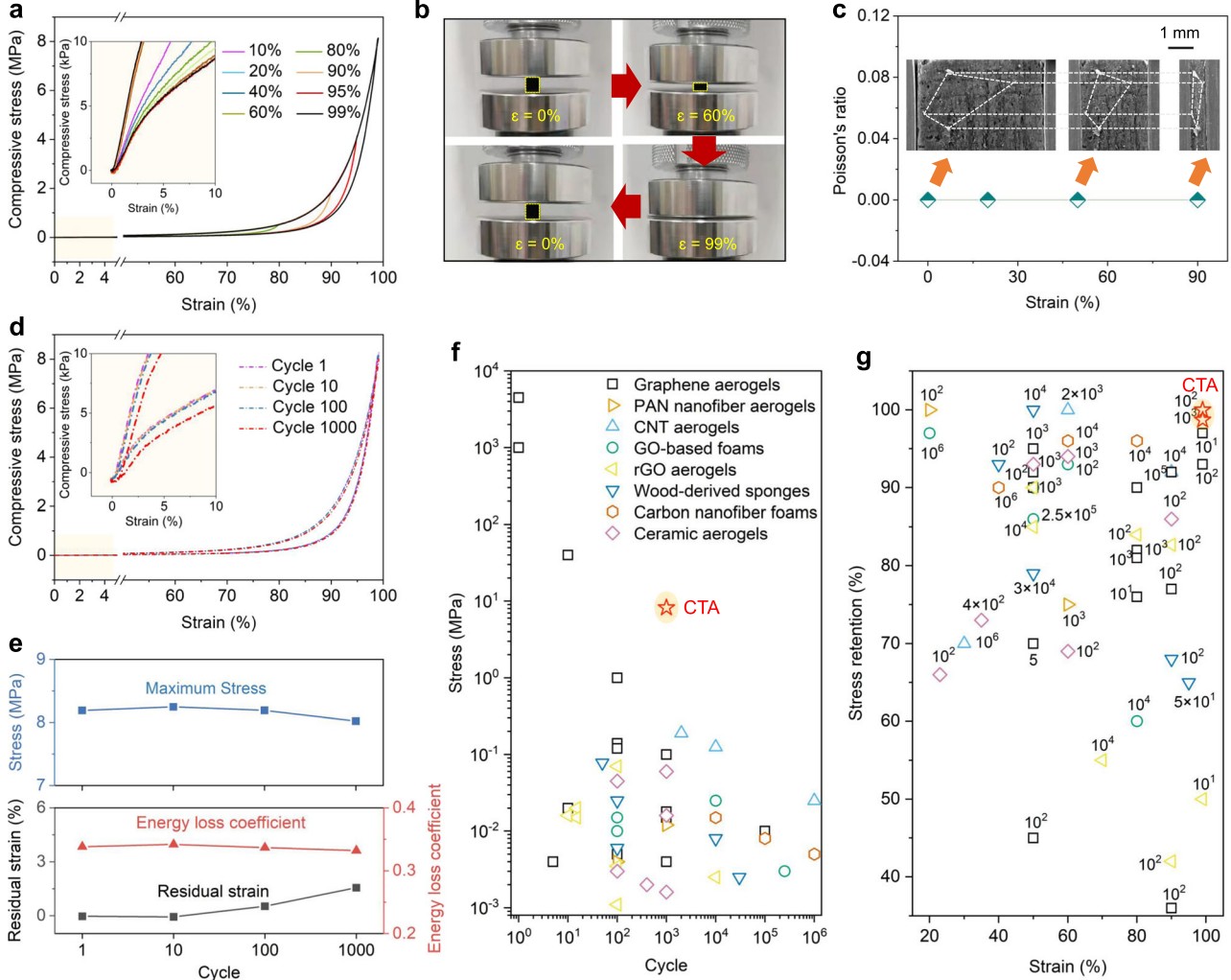

**Fig. 2 | Mechanical properties of carbon tube aerogels (CTAs). a** Stress vs. strain curves of a CTA (12.9 mg cm⁻³) compressed to 10–99% strain. Inset: Enlarging the onset of the compression curves, demonstrating no permanent deformation. **b** Experimental snapshots of a CTA (7 mm × 7 mm × 8 mm) under uniaxial compression to 99% strain. **c** Poisson's ratio of a CTA at different compressive strain (0–90%). Inset: Corresponding in situ scanning electron microscope (SEM) observations, showing a near-zero Poisson's ratio of the CTA. **d** Compressive stress vs. strain curves of a CTA at 99% strain for 1000 cycles. Inset: Enlarging the onset of the cyclic curves, showing that there is -1.5% residual deformation after 1000 cycles. **e** Residual strain, stress, and energy loss coefficient of CTAs for different compressive cycles. **f** Stress vs. cycle of CTAs compared to other elastic materials such as graphene aerogels[7–9,17,34–40], graphene oxide (GO)-based foams[10,11,41], polyacrylonitrile (PAN) and carbon nanofiber aerogels[12,42], wood-derived carbon sponges[43–45], carbon nanotube (CNT) aerogels[5,16], reduced graphene oxide (rGO) aerogels[13,46,47], and ceramic aerogels[18,40,48–52]. **g** Stress retention vs. strain of CTAs during cycles compared to previously reported materials[5,7–13,16–18,34–52]. The cycles are marked next to the symbols. Detailed information is available in Supplementary Table 1. Source data are provided as a Source Data file.

compressive resilience, and fatigue resistance, are one of the most elastic and durable, compared to other materials[5,7–13,16–18,34–52], as exhibited in Fig. 2f, g.

To investigate the thermal stability of CTAs, we subjected them to annealing at 2500 °C for 12 h in an Ar atmosphere. All samples remain structurally intact with no visible changes. We selected the CTA with the highest density of 43.2 mg cm⁻³ for further analysis. The original sample exhibits a compressive stress of 20.9 MPa at a recoverable strain of 98%, as shown in Supplementary Fig. 10a. Whereas after annealing, the compressive stress of the sample under the same testing conditions reduces to 11.9 MPa (see Supplementary Fig. 10b); however, interestingly, its elasticity appears to have slight improvement, as evident from a comparison of the curves in Supplementary Fig. 10c, d. Additionally, Supplementary Fig. 10e demonstrates that the annealed CTA presents a higher energy loss coefficient (an increase of 0.45–1 based on different strain), indicating an enhanced ability to absorb energy.

As presented in Supplementary Fig. 11, the disorder-induced *D* band (-1334 cm⁻¹) in CTAs weakens noticeably after 2500 °C annealing,

while the first-order graphite *G* band (-1582 cm⁻¹) becomes sharp and strong. This suggests a marked increase in intrinsic structural *sp²* order. The possible reason is that the ultra-high temperature annealing triggers *sp³*-to-*sp²* phase transition, causing carbon atoms to rearrange into a purely *sp²*-graphite structure owing to their high activity at 2500 °C. Our experimental results indicate that *sp³* may be one of the key factor to strengthen the carbon tubes, contributing to the improved mechanical properties of CTAs, particularly their stress values.

## Mechanism understanding of mechanical properties

To gain insight into the reversible compression mechanism of CTAs, we utilized in situ TEM to observe the structural evolution during the pressing and releasing process. As shown in Fig. 3a and Supplementary Movie 2, the tubes near the indenter initially move in the direction of uniaxial compression. With an increase in indentation depth, the loading force transfers to adjacent carbon tubes through joints, resulting in a larger range of carbon tubes deforming. This

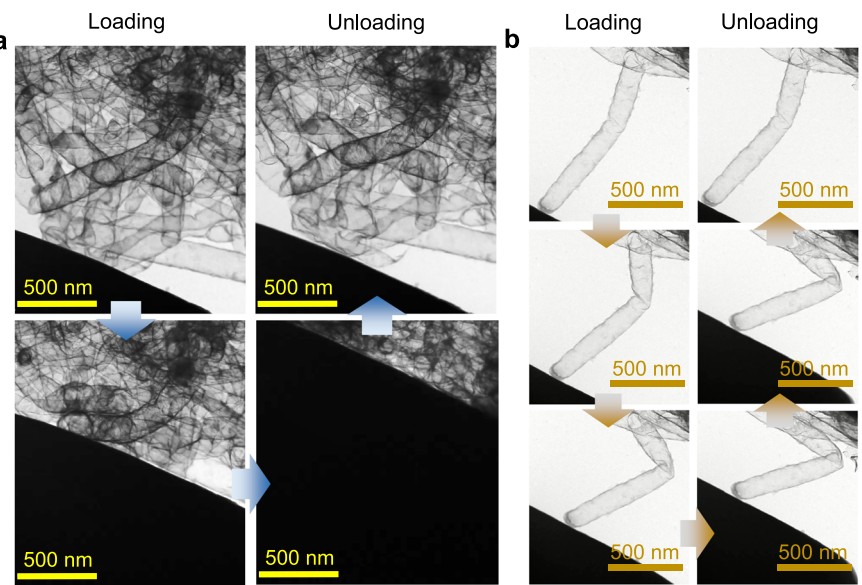

**Fig. 3 | Compressive mechanism of carbon tube aerogels (CTAs). a** In situ Transmission electron microscope (TEM) observations of carbon tube networks during uniaxial loading and unloading. **b** In situ TEM observations of a single carbon tube during bending.

phenomenon impedes local stress concentration and permanent damage, as the tubes can move to the most appropriate positions. This interpretation also explains the high mechanical values of CTAs, as presented in Fig. 2f, g. In situ TEM observations of a single tube (Fig. 3b and Supplementary Movie 3) confirm bending, rotating, and distorting of the carbon tube during compression. Upon release of the load, the compressed carbon tube springs back to its original shape, demonstrating good elasticity and flexibility.

To achieve deeper understanding of the atomic bending behaviors of carbon tubes, we carried out molecular dynamic (MD) analyses using Large-scale Atomic/Molecular Massively Parallel Simulator (LAMMPS) software[53]. It is well-known that the walls of purely $sp^2$ carbon tubes, where in-plane atoms are bonded through localized $sp^2$ hybridization, are connected by van der Waals forces between walls. As the tube bends, these weak interactions allow for relative motions such as wrinkling, sliding, and separating of the walls, as illustrated in Fig. 4a. Although these localized motions help prevent the breakage of $sp^2$ bonds (with a low breakage fraction of $sp^2$ bonds, at only 2.2%), they have a negative impact on the bending stability and overall strength of the tube.

When we introduced 5% $sp^3$ and established interwall coupling, as depicted in Fig. 4b, the shortened interwall distance and the pinning effect of $sp^3$ bonds between walls restrict the wall movements, resulting in minimal wrinkling in the $sp^2$-$sp^3$ hybrid tube. This improves the stiffness and strength of the tube, as demonstrated in Fig. 4c. Nevertheless, during bending of the tube, some $sp^3$ bonds may reach their maximum critical limits and separate (40.8% $sp^3$ in the present model, as shown in Fig. 4d). The breakage of these localized $sp^3$ bonds allows $sp^2$ walls to move freely again (see Fig. 4e and Supplementary Movie 4). Overall, the partial fracture of $sp^3$ bonds may not affect the elasticity of carbon tubes due to the majority of remaining $sp^2$ bonds.

**Dynamic mechanical and electromagnetic performance**

We conducted impact resistance tests on CTAs by dropping a steel ball that was ~200 times heavier than the samples. To capture the falling and rebounding process, we utilized a high-speed camera. As shown in Fig. 5a, when the ball hits the CTA surface at a velocity of 2083 mm s$^{-1}$, the sample undergoes significant deformation but quickly recovers without any structural damage or collapse. The rebound velocity of the ball is 847 mm s$^{-1}$. To calculate the impact and rebound energies, we used the formulas $E_i = mv^2_i/2$ and $E_r = mv^2_r/2$, where m represents the mass of the steel ball, $v_i$ is the impact velocity, and $v_r$ is the rebound velocity. The result shows that CTAs can dissipate 83.4% energy during impact testing, demonstrating that they may serve as a damping material.

We performed a thorough investigation into the viscoelastic properties of CTAs, utilizing a dynamic mechanical analysis (DMA) to analyze their storage modulus, loss modulus, and damping ratio. Figure 5b illustrates that CTAs present a rapid and precise dynamic response across a wide frequency range (0.1–150 Hz). The damping ratio remains consistent at an average of 0.25 throughout the tests, regardless of frequency. Moreover, CTAs display stable thermomechanical performance in air conditions within a broad temperature range of −120 to 300 °C. To assess their fatigue resistance, we subjected CTAs to 100,000 cycles at oscillatory $\varepsilon = 10\%$. The dynamic mechanical values for the 1st and 100,000th cycles show negligible changes, indicating that CTAs possess temperature-invariant elasticity and fatigue-resistant properties.

As CTAs are compressed, the carbon tubes come into close proximity, creating conductive channels that can increase their conductivity. For example, when the compressive strain reaches 95%, the σ of CTAs jumps from 0.12 to 111.69 S cm$^{-1}$, demonstrating a wide tuning range, as shown in Supplementary Fig. 12.

It is worth mentioning that, σ has a dramatic impact on the electromagnetic interference (EMI) shielding effectiveness (SE) of materials[53]. In this study, uncompressed CTAs in a low conductive state exhibit a reflection shielding effectiveness (SE$_R$) and an absorption shielding effectiveness (SE$_A$) of ~7 dB and ~8 dB, respectively, leading to a total shielding effectiveness (SE$_T$) of ~15 dB, as depicted in Fig. 5c.

Upon compressing the CTAs to $\varepsilon = 50\%$, a noticeable increase in SE$_T$ to ~45 dB is achieved. The proposed shielding mechanism of the compressed CTAs is illustrated in Fig. 5d. As subjected to compression, the carbon tubes are compelled to come into close contact, thereby facilitating the establishment of efficient electron transport channels. This enhances the migration and hopping of electrons. In addition, the laminar architecture of CTAs gives rise to the occurrence of multiple interfacial reflections, which enables electromagnetic waves to be reflected between the carbon tube layers until they are completely absorbed and dissipated as heat. For our laminar CTAs with a 2-mm thickness, numerous carbon tube layers can serve as effective barriers to electromagnetic waves, resulting in little transmission of waves through the CTAs.

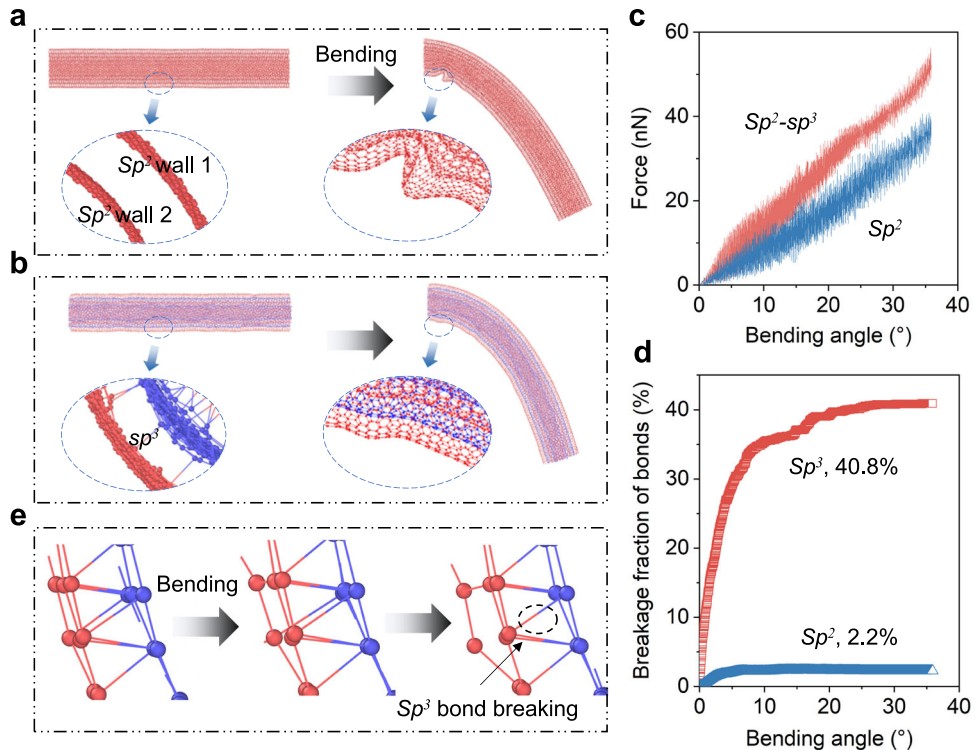

**Fig. 4 | Molecular dynamic (MD) simulation of five-layer carbon tubes during bending (simulated using LAMMPS software). a** The tube with purely $sp^2$ bonds. **b** The tube with $sp^2$-$sp^3$ hybrid bonds. **c** The force vs. bending angle curves of two types of carbon tubes during bending. **d** The breakage fraction of $sp^2$ and $sp^3$ bonds during bending. **e** Breaking process of $sp^3$ bonds during bending of a $sp^2$-$sp^3$ hybrid tube. Source data are provided as a Source Data file.

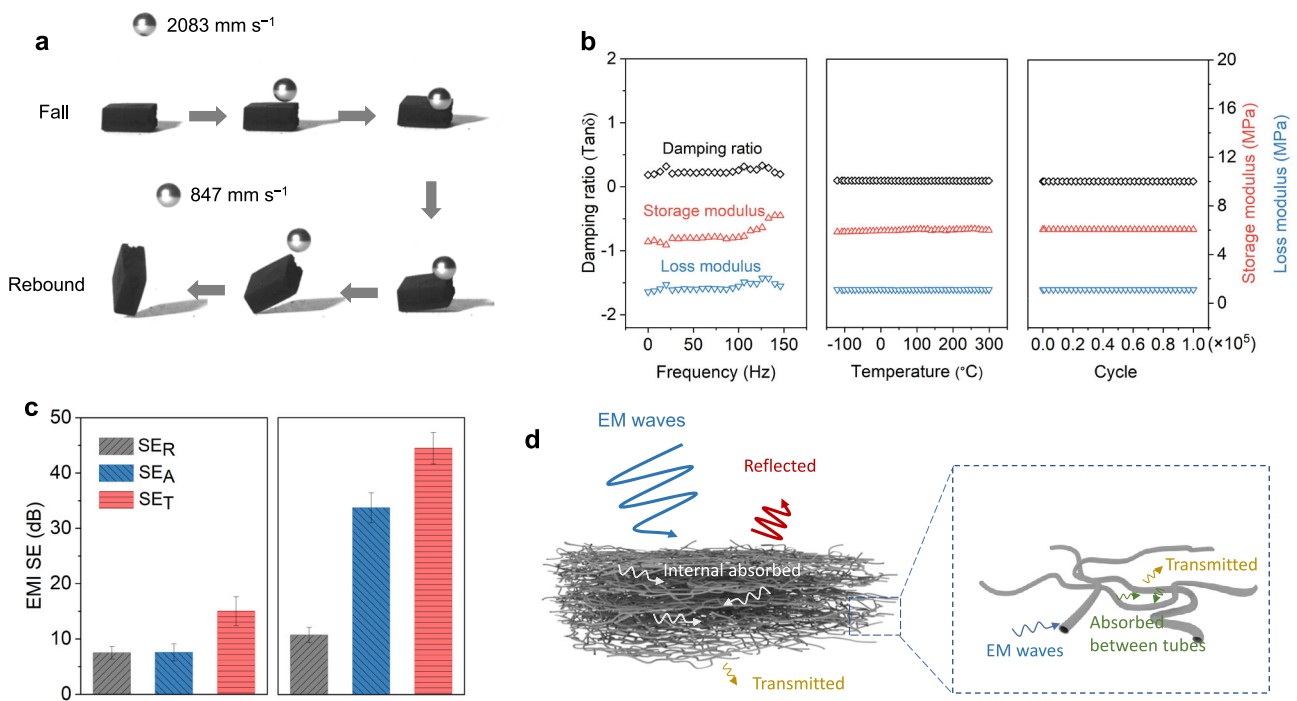

**Fig. 5 | Dynamic mechanical properties and electromagnetic interference (EMI) shielding performance of carbon tube aerogels (CTAs). a** Impact resistance of a CTA tested by dropping a free-falling steel ball. **b** Damping ratio, storage modulus, and loss modulus of CTAs measured by a dynamic mechanical analysis (DMA) at different frequencies, temperatures and compressive cycles. **c** EMI shielding effectiveness (SE) of CTAs at 0 and 50% strain. The reflection shielding effectiveness ($SE_R$), absorption shielding effectiveness ($SE_A$), and total shielding effectiveness ($SE_T$) for 5% compression are 7.48 ± 1.14, 7.54 ± 1.55, and 15.02 ± 2.61 dB, respectively, and are 10.75 ± 1.37, 33.73 ± 2.71, and 44.48 ± 2.85 dB, respectively, for 50% compression. **d** Schematic diagram of shielding mechanism of CTAs. Source data are provided as a Source Data file.

## Discussion

An effective approach was presented to fabricate high-performance CTAs. The highly interconnected carbon tubes with $sp^2$-$sp^3$ hybrid bonds enable rapid load transfer. As a result, the CTAs with a low density (-12–40 mg cm$^{-3}$) demonstrate good mechanical properties, including high mechanical strength (8.2–20.9 MPa), compressive resilience (up to 99% strain), and fatigue resistance (less than 1.5% permanent deformation after 1000 compressive cycles at 99% strain). Moreover, the CTAs possess high thermal stability up to 2500 °C in an Ar atmosphere.

Our CTAs exhibit promising potential as a viable candidate for various applications, such as impact resistance and energy dissipation, mainly since they can absorb up to 83.4% impact energy without incurring any structural damage, as demonstrated through free-falling steel ball testing. Furthermore, their high elasticity allows for adjustable electrical conductivity and tunable electromagnetic shielding behaviors. These properties make them ideal for use such as in wearable electronics, and high-precision pressure sensors.

## Methods

### Raw materials and chemicals

Methyltrimethoxysilane (MTMS, analytial reagent, 99.8%), dimethyl-dimethoxysilane (DMDMS, analytial reagent, 99.7%), deionized water (analytial reagent, 99%), and hydrofluoric acid (guaranteed reagent, 40%) were purchased from Meryer Technologies Co., Ltd. Please note that, it is extremely important to be fully aware of safety and emergency protocols when working with hydrofluoric acid. Ethanol (analytial reagent, 99.5%) was purchased from Sinopharm Chemical Reagents Co., Ltd. Short carbon fibers (powders, 99%) were purchased from Guangwei Composites Co., Ltd. Hydrogen (H$_2$, 99.999%), argon (Ar, 99.999%), and methane (CH$_4$, 99.999%) gases were purchased from Xi'an Jiayihe Gas Co., Ltd. All chemicals were used in their as-received conditions without further purification.

### Synthesis of SiC nanowire aerogels

MTMS (40 g), DMDMS (10 g), ethanol (100 g), and deionized water (60 g) were mixed by mechanical stirring for 30 min to form an ethanol solution of siloxane sol[27]. The hydrolysis of -Si-OCH$_3$ group from MTMS and DMDMS, results in the formation of -Si-OH, while the polycondensation between -Si-OH groups leads to the formation of siloxane sol. Short carbon fibers (20 g) were put and dispersed in the siloxane sol by mechanical agitation for 20–30 min to form a mixture solution.

The mixture solution was subjected to vacuum filtration to remove the liquids, and the carbon fiber skeletons with viscous siloxane sol left in the pores were obtained. the skeletons were then treated at 100 °C in an oven for 5 h. Subsequently, the skeletons with siloxane gel were heated to 1550 °C at a rate of 5 °C min$^{-1}$ in an Ar atmosphere and a constant pressure of 0.2 MPa. After holding for 2 h in this condition, entangled SiC nanowires were formed inside the skeletons according to the following reactions:

$$SiO(s) + 2C(s) \rightarrow SiC(s) + CO(g) \qquad (1)$$

$$SiO(s) + CO(g) \rightarrow SiC(s) + CO_2(g) \qquad (2)$$

The obtained skeletons were treated at 700 °C in air for 3 h to remove the carbon fibers. Finally, high-purity SiC nanowire aerogels with a density of 10–20 mg cm$^{-3}$ were acquired. The yield of isolated mass for SiC nanowire aerogels (the mass ratio of nanowires to siloxane sol) were calculated to be -40%.

### Synthesis of laminar SiO$_2$ nanowire aerogels

Hot-pressing (1200 °C for 2 h) was applied on the raw SiC nanowire aerogels to increase their density. The pressure was set to be 10–15 MPa according to the desired density of nanowires (200–300 mg cm$^{-3}$). During the hot-pressing process, a laminar structure naturally forms in the aerogels. Then, these aerogels were subjected to 1000 °C in a furnace for 24 h to fully oxidize SiC into SiO$_2$. As a result, laminar SiO$_2$ nanowire aerogels were obtained.

### Synthesis of CTAs

The SiO$_2$ laminar nanowire aerogels were to deposit carbon layers by CVD. The CVD temperature was set to be 1190 °C, and CH$_4$ (20 ml min$^{-1}$), H$_2$ (50 ml min$^{-1}$), and Ar (200 ml min$^{-1}$) were used as reactive gas, purify gas and dilute gas, respectively. The deposition time was 70–280 min according to the final density of CTAs (-12–40 mg cm$^{-3}$).

The SiO$_2$ nanowire cores were completely etched by hydrofluoric acid for 6 h at 60 °C in air conditions, followed by washing with deionized water for several times. When handling hazardous hydrofluoric acid, independent of the concentrations, it is critical to be aware of the risk assessment and requires safety protocols. The wet CTAs were naturally dried for 24 h, and ultimately hollow CTAs (with a density of -12–40 mg cm$^{-3}$) were acquired.

### Material characterization

The microstructure evolution of CTAs was investigated using in situ SEM (Quanta 600, FEI, United States) operated at an acceleration voltage of 200 kV, and in situ TEM (JEM-2100, JEOL, Japan) equipped with EDS at an acceleration voltage of 200 kV. The elemental and structural analyses were carried out by XPS (ESCALAB Xi+, Thermo Fisher, United States), Raman (InVia Qontor, Renishaw, United States) with a 532-nm laser, and aberration-corrected TEM (HF5000, Hitachi, Japan) at an acceleration voltage of 100 kV equipped with EELS. The band ratio of $\pi^*$ to ($\sigma^* + \pi^*$) was determined by integrating the $\pi^*$ peak ($I_\pi$) and the remaining area starting at the $\sigma^*$ edge ($I_o$) using a two-window method[32]. The compressive properties of CTAs were investigated via a universal tester (UTM2103, Suns Co., Ltd., China) with a 1000 N load cell. The loading and unloading rates were 5 mm min$^{-1}$. The viscoelastic properties of CTAs were studied using DMA (DMA242E, Netzsch, Germany). At least 5 samples were tested for each condition, and no data were excluded from the analyses.

EMI shielding tests were performed on a vector network analyzer (E5071C, Rohde & Schwarz, Germany). The samples were cut into a cuboid shape (22.84 mm × 10.14 mm × 2 mm). The SE$_R$ and SE$_A$ were be expressed as[54]:

$$SE_R = 20\log\left(\frac{\sqrt{u_0\sigma}}{4\sqrt{2\pi f u_\varepsilon}}\right) \qquad (3)$$

$$SE_A = 8.686d\sqrt{\pi f u\sigma} \qquad (4)$$

where f is the testing frequency, $u$ is the magnetic permeability of a material, $u_0$ and $\varepsilon_0$ are the permeability and permittivity of free space, respectively, and $d$ is the thickness of a material.

Experimentally, SE$_R$ and SE$_A$ were measured in terms of reflection and effective absorption, considering the power of the incident electromagnetic waves inside a shielding material as[55]:

$$SE_R = 10\log\left(\frac{1}{1-|S11|^2}\right) \qquad (5)$$

$$SE_A = 10\log\left(\frac{1-|S11|^2}{|S21|^2}\right) \qquad (6)$$

where S11 and S21 are the scattering parameters given by the vector network analyzer directly. The $SE_T$ was given as:

$$SE_T = SE_R + SE_A \tag{7}$$

At least 3 samples were tested for EMI shielding tests, and no data were excluded from the analyses. The data presented in the article were the mean values of all samples.

## Computational simulations

LAMMPS software was utilized to conduct MD simulations on carbon tubes. The five-layer tubes were constructed at a $L/D$ ratio of 6.2. To create the $sp^2$-$sp^3$ hybrid tube, 5% $sp^3$ was set to be randomly distributed among the remaining 95% $sp^2$ bonds. The purely $sp^2$ tube was created by utilizing exclusively $sp^2$ bonds. Upon bending, a curvature of 35.9° was given. In general, the bond lengths of $sp^2$ and $sp^3$ in carbon atoms fell within the range of 0.13–0.15 and 0.15–0.17 nm, respectively[56–58]. Therefore, we defined when $sp^2$ and $sp^3$ bonds exceeded 0.16 and 0.18 nm, respectively, they were considered broken. The visualization of carbon tube models was conducted using OVITO[59].

## Data availability

The data that supports the findings of this study are available in the article and supplementary information file. Source data are provided with this manuscript. Source data are provided with this paper.

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

## Acknowledgements

The authors acknowledge the financial support from the Technical Field Foundation of China and National Natural Science Foundation of China (No. 52102077, 52072294, and 92263204). Additionally, the authors would like to thank the Instrument Analysis Center of Xi'an Jiaotong University for the assistance with TEM, XPS, and Raman tests, as well as W.L, T.Y.L., and H.Y.N. at Northwestern Polytechnical University for the help with high-temperature annealing and EMI shielding tests.

## Author contributions

L.Z. and H.J.W. conceived and designed the research. L.Z., D.L., and J.J.Z. participated in materials preparation and mechanical tests. L.Z. wrote the paper. H.J.W. supervised the research. Y.B.Q. and P.Z. carried out the in situ TEM characterization. P.F.G., L.S., X.L., M.N., and K.P. discussed the results and analyzed the data.

## Competing interests

The authors declare no competing interests.
