## [Peer Review File · Nature Communications]

Reviewer comments, first round –

Reviewer #1 (Remarks to the Author):

The authors report on the synthesis of elastic carbon tube aerogels with high compressive resilience. The reported synthesis method is based on the conformal coating of carbon on SiO₂ nanowires using chemical vapor deposition followed by the HF removal of the SiO₂ cores. The resulting carbon tube aerogels consist of a network of interconnected hollow carbon tubes. The authors demonstrated that the carbon tubes are highly compressible and fatigue resistant. They also possess variable electrical conductivity and electromagnetic shielding effectiveness. Overall, it is an interesting study that shows an innovative synthesis method of highly elastic materials with unique properties. However, there are several issues that should be clarified before I can come to a final recommendation.

1. The title is misleading. The reported materials contain no rubber. There are purely carbon-based elastomers. The term "rubber" should not be used. I also do not understand the term "interwall" in the title as there are no walls in the materials, but rather "interconnected tubes". The authors need to modify the title to correctly represent the main findings of the paper.

2. The authors claim that their carbon aerogels can withstand the highest compressive stress of 20.9 MPa, which is not true. There have been at least two reports on graphene aerogels in the literature which have reported a much higher maximum compressive stress, e.g. 1 GPa in the report of C. Li et al., *Nanoscale* 10, 18291–18299 (2018) and 4.5 GPa in the report of M. Silhavić et al., *Communications Physics* (2022) 5:27. These reports are missing in Fig. 2(f) where the comparison to the other elastic materials is shown. The authors need to correct it and properly list all the relevant literature in the field.

3. The authors claim that the carbon tubes are strongly bridged by sp³ bonds and that these bonds are responsible for the improved stiffness. However, the presence of the sp³ bridging between individual tubes is not demonstrated experimentally. The Raman and EELS spectra show the sp² character of the carbon material with many sp³ defects. EELS mapping also shows the homogenous distribution of π^* and σ^* bands along the tube. The authors do not provide proof of the increased sp³ content at the joint between two tubes, which means that sp² bonding might still be dominant at the joints and responsible for the observed mechanical properties. The authors should provide better experimental proof or change their claims and interpretation of the experimental data in the manuscript.

4. The interpretation of the EELS spectra shown in Figures 1(h) and S5 is oversimplified. It cannot give any quantitative analysis of sp² and sp³ content in the material using the "two-window intensity-ratio" method. It is very difficult to quantitatively determine the sp² and sp³ fractions of unknown carbon materials from electron energy-loss near-edge structure analysis of C-K spectra, see e.g. M. Diociaiuti et al., *Micron*, 90, (2016) 97-107. The EELS mapping shows only the ratio between states in the π^* band and the σ^* band and does not show the ratio of the sp² to sp³ content. Both the π^* and σ^* bands in the EELS spectra in Figure seem to be of sp² character. In sp² hybridized carbon, each threefold (or sp²) site contributes one state to the π^* band and three states to the σ^* band, while each fourfold (or sp³) site contributes a total of four states to the σ^* band. On the other hand, the EELS spectra of pure sp³ material should not contain any π^* band, and the energy loss and shape of the σ^* band should be significantly different. The authors need to correct the EELS data analysis and properly interpret it in the paper.

5. There is a typo in the last sentence of the abstract on Page 1, line 20. The term "used" does not fit the context. The authors should correct it.

6. There is a missing description of the two TEM figures on the right in Figure 1(d).

7. There is a wrong scale bar in the atomic resolution TEM image of carbon tubes in Figure 1(e). It shows 10 micrometers. Moreover, the TEM image shows a very strange atomic structure for

carbon materials. The atomic structure in the TEM image looks more like SiC than carbon. The authors should provide TEM images with correct scale bars and show diffraction patterns from TEM or XRD to confirm the carbon structure. This is a very important point that can change the overall interpretation of the observed mechanical properties.

8. The SEM images in the inset of Figure 2(c) do not correspond to the strain shown in the plot below. The SEM shows much lower strain values. The authors should display the corresponding images of the deformed samples.

9. Explaining the mechanism of the high mechanical properties using molecular dynamics simulation is interesting. However, it is not clear how the sp² and sp³ bonds have been modeled in the MD simulation. The authors should show better images of the atomic models of the structures and show how purely sp² and sp² with 5% of sp³ bonds differ in terms of bond breakage. The modeling also seems to consider the breakage of sp³ bonds only, which is not reflecting the experimental condition where a majority of sp² bonds is found. Both sp² and sp³ breakage should be considered. The authors should also discuss in more detail the role of the local wrinkling in these two scenarios on the bending stability of the tubes. This will be very useful for the research community.

Reviewer #2 (Remarks to the Author):

A simple approach combining template sacrificing and chemical vapor deposition (CVD) was developed to prepare CTAs. They used silicon-carbide (SiC) nanowire aerogels as raw template materials. The template plus CVD method is similar to previous publications about the carbon tube network preparation. It is really costly.

The in situ TEM observation of carbon tubes during uniaxial loading and unloading is not clear. The mess cross linked tubes do not mean the statement. The observation of a single tube is recommended.

Cross linked point of carbon tube should be characterized in detail.

More evidence for the sp² fraction and sp³ fraction should be provided. EELS mapping of carbon tubes in this work could not give a strong support for the sp² fraction.

It is no big meaning to compare the ultimate stress versus maximum strain of CTAs compared to other elastic materials. They should compare the stress versus at a different certain strain of CTAs corresponding to other elastic materials.

REVIEWER COMMENTS

Title: Carbon tube aerogels with high cross-links for super-elasticity and fatigue resistance

Manuscript ID: NCOMMS-23-01579-A

Reviewer #1 (Remarks to the Author):

The authors report on the synthesis of elastic carbon tube aerogels with high compressive resilience. The reported synthesis method is based on the conformal coating of carbon on SiO₂ nanowires using chemical vapor deposition followed by the HF removal of the SiO₂ cores. The resulting carbon tube aerogels consist of a network of interconnected hollow carbon tubes. The authors demonstrated that the carbon tubes are highly compressible and fatigue resistant. They also possess variable electrical conductivity and electromagnetic shielding effectiveness. Overall, it is an interesting study that shows an innovative synthesis method of highly elastic materials with unique properties. However, there are several issues that should be clarified before I can come to a final recommendation.

1. The title is misleading. The reported materials contain no rubber. There are purely carbon-based elastomers. The term “rubber” should not be used. I also do not understand the term “interwall” in the title as there are no walls in the materials, but rather “interconnected tubes”. The authors need to modify the title to correctly represent the main findings of the paper.

2. The authors claim that their carbon aerogels can withstand the highest compressive stress of 20.9 MPa, which is not true. There have been at least two reports on graphene aerogels in the literature which have reported a much higher maximum compressive stress, e.g. 1 GPa in the report of C. Li et al., *Nanoscale* 10, 18291–18299 (2018) and 4.5 GPa in the report of M. Silhavić et al., *Communications Physics* (2022) 5:27. These reports are missing in Fig. 2(f) where the comparison to the other elastic materials is shown. The authors need to correct it and properly list all the relevant literature in the field.

3. The authors claim that the carbon tubes are strongly bridged by sp³ bonds and that these bonds are responsible for the improved stiffness. However, the presence of the sp³ bridging between individual tubes is not demonstrated experimentally. The Raman and EELS spectra show the sp² character of the carbon material with many sp³ defects. EELS mapping also shows the homogenous distribution of π* and σ* bands along the tube. The

authors do not provide proof of the increased sp^3 content at the joint between two tubes, which means that sp^2 bonding might still be dominant at the joints and responsible for the observed mechanical properties. The authors should provide better experimental proof or change their claims and interpretation of the experimental data in the manuscript.

4. The interpretation of the EELS spectra shown in Figures 1(h) and S5 is oversimplified. It cannot give any quantitative analysis of sp^2 and sp^3 content in the material using the “two-window intensity-ratio” method. It is very difficult to quantitatively determine the sp^2 and sp^3 fractions of unknown carbon materials from electron energy-loss near-edge structure analysis of C-K spectra, see e.g. M. Diociaiuti et al., *Micron*, 90, (2016) 97-107. The EELS mapping shows only the ratio between states in the π^* band and the σ^* band and does not show the ratio of the sp^2 to sp^3 content. Both the π^* and σ^* bands in the EELS spectra in Figure seem to be of sp^2 character. In sp^2 hybridized carbon, each threefold (or sp^2) site contributes one state to the π^* band and three states to the σ^* band, while each fourfold (or sp^3) site contributes a total of four states to the σ^* band. On the other hand, the EELS spectra of pure sp^3 material should not contain any π^* band, and the energy loss and shape of the σ^* band should be significantly different. The authors need to correct the EELS data analysis and properly interpret it in the paper.

5. There is a typo in the last sentence of the abstract on Page 1, line 20. The term “used” does not fit the context. The authors should correct it.

6. There is a missing description of the two TEM figures on the right in Figure 1(d).

7. There is a wrong scale bar in the atomic resolution TEM image of carbon tubes in Figure 1(e). It shows 10 micrometers. Moreover, the TEM image shows a very strange atomic structure for carbon materials. The atomic structure in the TEM image looks more like SiC than carbon. The authors should provide TEM images with correct scale bars and show diffraction patterns from TEM or XRD to confirm the carbon structure. This is a very important point that can change the overall interpretation of the observed mechanical properties.

8. The SEM images in the inset of Figure 2(c) do not correspond to the strain shown in the plot below. The SEM shows much lower strain values. The authors should display the corresponding images of the deformed samples.

9. Explaining the mechanism of the high mechanical properties using molecular dynamics

simulation is interesting. However, it is not clear how the sp^2 and sp^3 bonds have been modeled in the MD simulation. The authors should show better images of the atomic models of the structures and show how purely sp^2 and sp^2 with 5% of sp^3 bonds differ in terms of bond breakage. The modeling also seems to consider the breakage of sp^3 bonds only, which is not reflecting the experimental condition where a majority of sp^2 bonds is found. Both sp^2 and sp^3 breakage should be considered. The authors should also discuss in more detail the role of the local wrinkling in these two scenarios on the bending stability of the tubes. This will be very useful for the research community.

Reviewer #2 (Remarks to the Author):

A simple approach combining template sacrificing and chemical vapor deposition (CVD) was developed to prepare CTAs. They used silicon-carbide (SiC) nanowire aerogels as raw template materials. The template plus CVD method is similar to previous publications about the carbon tube network preparation. It is really costly.

The in situ TEM observation of carbon tubes during uniaxial loading and unloading is not clear. The mess cross linked tubes do not mean the statement. The observation of a single tube is recommended.

Cross linked point of carbon tube should be characterized in detail.

More evidence for the sp^2 fraction and sp^3 fraction should be provided. EELS mapping of carbon tubes in this work could not give a strong support for the sp^2 fraction.

It is no big meaning to compare the ultimate stress versus maximum strain of CTAs compared to other elastic materials. They should compare the stress versus at a different certain strain of CTAs corresponding to other elastic materials.

Point-by-Point Response to the Reviewers' Comments

Dear editor and reviewers,

We are glad to adopt your suggestions for revising our paper entitled “**Carbon tube aerogels with high cross-links for super-elasticity and fatigue resistance**”. These comments are very valuable and helpful for improving our manuscript. According to the comments, we have tried our best to revise the manuscript. We hope that the corrections will meet with approval. Revised portion was marked in red in the revised manuscript. The detailed responses are listed as follow:

Reviewer #1 (Remarks to the Author):

The authors report on the synthesis of elastic carbon tube aerogels with high compressive resilience. The reported synthesis method is based on the conformal coating of carbon on SiO₂ nanowires using chemical vapor deposition followed by the HF removal of the SiO₂ cores. The resulting carbon tube aerogels consist of a network of interconnected hollow carbon tubes. The authors demonstrated that the carbon tubes are highly compressible and fatigue resistant. They also possess variable electrical conductivity and electromagnetic shielding effectiveness. Overall, it is an interesting study that shows an innovative synthesis method of highly elastic materials with unique properties. However, there are several issues that should be clarified before I can come to a final recommendation.

Thank you for this summary of our research.

1. The title is misleading. The reported materials contain no rubber. There are purely carbon-based elastomers. The term “rubber” should not be used. I also do not understand the term “interwall” in the title as there are no walls in the materials, but rather “interconnected tubes”. The authors need to modify the title to correctly represent the main findings of the paper.

Thanks for your kind suggestion. In order to avoid misleading, we have revised our title to “Carbon tube aerogels with high cross-links for super-elasticity and fatigue resistance” (high cross-links have the meaning of interconnected tubes).

2. The authors claim that their carbon aerogels can withstand the highest compressive stress of 20.9 MPa, which is not true. There have been at least two reports on graphene

aerogels in the literature which have reported a much higher maximum compressive stress, e.g. 1 GPa in the report of C. Li et al., *Nanoscale* 10, 18291–18299 (2018) and 4.5 GPa in the report of M. Silhvik et al., *Communications Physics* (2022) 5:27. These reports are missing in Fig. 2(f) where the comparison to the other elastic materials is shown. The authors need to correct it and properly list all the relevant literature in the field.

Thank you for your reminding. We have added more relevant literature (including the two papers you mentioned) in Fig. 2f and g, and also in Supplementary Table 1.

Fig. 2 Mechanical properties of CTAs. (a) Compression curves of CTAs ($\sim 12 \text{ mg cm}^{-3}$) with strain ranging from 10–99%. Inset: Enlarging the onset of compression curve, indicating no permanent deformation. (b) Experimental snapshots of CTAs under uniaxial compression to 99% strain. (c) Near-zero ν of CTAs at different compressive strain (0–90%). Inset shows corresponding *in situ* SEM observation. (d) Compressive stress–strain curves of CTAs at 99% strain for 1,000 cycles. (e) Residual strain, stress, and energy loss coefficient of CTAs for different compressive cycles. (f) Compression cycle versus strain of CTAs compared to other elastic materials such as graphene aerogels,^{7-9,17,34,35,38,39,46-48} fiber aerogels,^{12,40} wood-derived sponges,⁴¹⁻⁴³ carbon nanotube aerogels,^{5,16} GO foams,^{10,11,44}, and ceramic aerogels.^{18,48-53} (g) Compressive strain versus stress retention of CTAs during cycles compared to previously reported materials. The cycle numbers are marked next to the symbols.^{5,7-13,16-18,34-53} Detailed information is shown in Supplementary Table 1.

3. The authors claim that the carbon tubes are strongly bridged by sp^3 bonds and that these bonds are responsible for the improved stiffness. However, the presence of the sp^3 bridging between individual tubes is not demonstrated experimentally. The Raman and ELLS

spectra show the sp^2 character of the carbon material with many sp^3 defects. EELS mapping also shows the homogenous distribution of π^* and σ^* bands along the tube. The authors do not provide proof of the increased sp^3 content at the joint between two tubes, which means that sp^2 bonding might still be dominant at the joints and responsible for the observed mechanical properties. The authors should provide better experimental proof or change their claims and interpretation of the experimental data in the manuscript.

Sorry for the misleading. We did not mean the carbon tubes are bridged by sp^3 bonds. In fact, as mentioned by the reviewer, the distribution of π^* and σ^* bands along the tube is homogenous. We just want to express that, there are sp^3 bonds in the walls of carbon tubes which are attributed to the improved stiffness compared to purely sp^2 carbon tubes. In order to avoid ambiguity, we have decided to change our description of “carbon tubes bridged by sp^3 bonds” to “the tube contains both sp^2 and sp^3 ” in the manuscript.

4. The interpretation of the EELS spectra shown in Figures 1(h) and S5 is oversimplified. It cannot give any quantitative analysis of sp^2 and sp^3 content in the material using the “two-window intensity-ratio” method. It is very difficult to quantitatively determine the sp^2 and sp^3 fractions of unknown carbon materials from electron energy-loss near-edge structure analysis of C-K spectra, see e.g. M. Diociaiuti et al., *Micron*, 90, (2016) 97-107. The EELS mapping shows only the ratio between states in the π^* band and the σ^* band and does not show the ratio of the sp^2 to sp^3 content. Both the π^* and σ^* bands in the EELS spectra in Figure seem to be of sp^2 character. In sp^2 hybridized carbon, each threefold (or sp^2) site contributes one state to the π^* band and three states to the σ^* band, while each fourfold (or sp^3) site contributes a total of four states to the σ^* band. On the other hand, the EELS spectra of pure sp^3 material should not contain any π^* band, and the energy loss and shape of the σ^* band should be significantly different. The authors need to correct the EELS data analysis and properly interpret it in the paper.

Thanks for your comment. We agree with you that it is very difficult to precisely and quantitatively determine the sp^2 and sp^3 fractions up to now. However, from EELS mapping, it is certain that the ratio of π^* to σ^* bands varies in the tubes. Please see Supplementary Fig. 5 where spectrum curves drew from EELS pixels show different peak intensity of π^* and σ^* bands.

According to your comment, we have revised and added our statement as below in the manuscript: It is noteworthy that, these sp^2 and sp^3 of carbon systems correspond to very defined orbital states.³² Sp^2 is the hybridization of the atomic $2s$ orbital with $2p_x$ and $2p_y$ orbitals, resulting in a system of three planar σ -bonds forming an angle of 120° . The remaining $2p_z$ orbital that is perpendicular to the plane of the σ -bonds, forms a π -bond. Sp^3 corresponds to the hybridization of the atomic $2s$ orbital with its three $2p_x$, $2p_y$ and $2p_z$ orbitals in equal proportion

and results in a system of four σ -bonds at an angle of 109.5° .³³ Related to these bonding states are the unoccupied anti-bonding states σ^* and π^* . As can be seen in Fig. 1h, electron energy loss spectroscopy (EELS) mapping shows that, the π^* fraction varies between 0.32–0.45 and the distribution of π^* and σ^* bands along the tube are roughly homogenous (selected typical raw curves of C-K spectra are given in Supplementary Fig. 5). That might suggest a uniform mix of sp^2 and sp^3 bonds in the walls of tubes.

Supplementary Fig. 5. Typical EELS spectra obtained from a single carbon tube. (a) EELS mapping with pixels that shows the ratio of π^* to σ^* using “two-window intensity-ratio” method. (b) Raw curve data drew from the selected pixels, which show different peak intensity of π^* and σ^* .

5. There is a typo in the last sentence of the abstract on Page 1, line 20. The term “used” does not fit the context. The authors should correct it.

We have corrected the term in the manuscript.

6. There is a missing description of the two TEM figures on the right in Figure 1(d).

Thanks for your reminding. The top right image is the dark field TEM image of carbon tubes, and the bottom right one is the corresponding energy diffraction pattern of the C element in the network. We have added the labels in Fig. 1d.

7. There is a wrong scale bar in the atomic resolution TEM image of carbon tubes in Figure 1(e). It shows 10 micrometers. Moreover, the TEM image shows a very strange atomic structure for carbon materials. The atomic structure in the TEM image looks more like SiC than carbon. The authors should provide TEM images with correct scale bars and show diffraction patterns from TEM or XRD to confirm the carbon structure. This is a very important point that can change the overall interpretation of the observed mechanical properties.

Sorry for our mistake. We have remade the scale bar for Fig.1e. The TEM image in Fig. 1e was extracted from Fig. R1. As can be found below, the wall of carbon tubes consists of dozens of curved layers.

Fig. R1. Low- (left) and high-resolution (right) TEM images of the wall of a carbon tube.

As for single-crystal SiC, its typical TEM image is provided in Fig. R2. Obviously, it has totally different structure compared to Fig. R1, and its atoms are much more ordered.

Fig. R2. Typical TEM image of a single-crystal SiC nanowire.

In addition, to further confirm whether there is Si element in our materials, we carried out EDS and EELS analysis from TEM. As seen in Fig. R3, the tubes contain only C elements according to EDS mapping. And, there are missing characteristic peaks of Si element at about 90 and 1,950 eV based on EELS results from TEM. From the above analysis, we draw a conclusion the materials consist of only carbon. We have put Fig. R3 into the Supplementary Materials (Supplementary Fig. 6) and added the corresponding interpretation in the manuscript: TEM images and corresponding elemental analysis of the joints between carbon tubes are shown in Supplementary Fig. 6. From EELS spectra, we confirm that SiO₂ has been removed out and no Si element is left in the tubes.

Fig. R3. Microstructure of the joint between two carbon tubes. (a) TEM image of the tubes. (b) EDS mapping of the tubes. (c) EELS spectrum curves of a single tube with energy loss ranging from 50–150 eV and 1850–2000 eV respectively, which suggest no Si element in the tubes.

8. The SEM images in the inset of Figure 2(c) do not correspond to the strain shown in the plot below. The SEM shows much lower strain values. The authors should display the corresponding images of the deformed samples.

We have retested the compression of CTAs using mechanical testing system equipped with *in-situ* SEM. The results can be seen below and in Fig. 2c.

Fig. R4. SEM images of a CTA compressed to different strain. (a) 0%. (b) ~48%. (c) ~88.5%.

9. Explaining the mechanism of the high mechanical properties using molecular dynamics simulation is interesting. However, it is not clear how the sp^2 and sp^3 bonds have been modeled in the MD simulation. The authors should show better images of the atomic models of the structures and show how purely sp^2 and sp^2 with 5% of sp^3 bonds differ in terms of bond breakage. The modeling also seems to consider the breakage of sp^3 bonds only, which is not reflecting the experimental condition where a majority of sp^2 bonds is found. Both sp^2 and sp^3 breakage should be considered. The authors should also discuss in more detail the role of the local wrinkling in these two scenarios on the bending stability of the tubes. This will be very useful for the research community.

Thank you for your comment. In fact, we have considered also the breakage of sp^2 in 5-wall carbon tubes. The breakage fraction of sp^3 was measured to be about 38% while that of sp^2 was only about 2% (so we did not interpret this part in the previous version of manuscript). We now provide breakage fractions of both sp^2 and sp^3 in Fig. 3e. Such small breakage numbers of sp^2 should be related to the relatively localized motions in purely sp^2 -hybridized tube during bending (see Supplementary Fig. 12). In other words, since there are only weak van der Waals forces between walls of a sp^2 carbon tube, relative motions between walls (such as wrinkling, sliding, and separating as shown in Supplementary Fig. 12a) can easily occur. These motions can avoid the breakage of sp^2 bonds, but also mean the purely sp^2 carbon tube seems more like a “soft” material with relatively low stiffness and bending stability.

When we added 5% sp^3 bonds between sp^2 walls of a carbon tube, connections between sp^2 -walls were established (see Fig.3c). The relative motions between walls are restricted to some extent and thus little wrinkling happens in the sp^2 - sp^3 hybrid tube (Supplementary Fig. 12b). This can improve the stiffness and strength of a tube (see Supplementary Fig. 13). But upon bending of a tube, it is certain that some sp^3 bonds reaching their maximum critical limits, will break (38.8% sp^3 in the present model, as seen in Fig. 3e). The cleavage of those localized sp^3 bonds makes sp^2 -walls to move freely again, as shown in Fig. 3d and Supplementary Movie 4. Overall, the partial fracture of sp^3 bonds might not affect the elasticity of carbon tubes due to the remaining of the majority of sp^2 bonds.

We have added the above discussion into the manuscript.

Fig. 3 Compressive mechanism of CTAs. (a) *in situ* TEM observation of carbon tubes during uniaxial loading and unloading. (b) *In situ* TEM observation of a single carbon tube during bending. (c) Molecular dynamic simulation of a five-layer carbon tube with sp^2 - sp^3 hybrid bonds during bending using Lammps software. (d) Breaking process of sp^3 bonds during bending of carbon tube. (e) breakage fractions of sp^2 and sp^3 bonds during bending.

Supplementary Fig. 12. Molecular dynamic simulation of carbon tubes using Lammmps.

(a) The tube with purely sp^2 bonds (b) The tube with hybridized sp^2-sp^3 bonds.

Reviewer #2 (Remarks to the Author):

A simple approach combining template sacrificing and chemical vapor deposition (CVD) was developed to prepare CTAs. They used silicon-carbide (SiC) nanowire aerogels as raw template materials. The template plus CVD method is similar to previous publications about the carbon tube network preparation. It is really costly.

Thanks for your summary. As we know, cross-links between struts are the key for the mechanical performance of a material. In the present work, we propose a new way to prepare carbon tube aerogels with much higher cross-links over other work, and thus result in improved mechanical properties.

1. The in situ TEM observation of carbon tubes during uniaxial loading and unloading is not clear. The mess cross linked tubes do not mean the statement. The observation of a single tube is recommended.

Thank you for your good suggestion. We have added the TEM observation of bending a tube into Fig. 3b and provided corresponding movie (see Supplementary Movie 3). More discussion for these images are given: *In situ* TEM observation of a single tube in Fig. 3b and Supplementary Movie 3 confirm bending, rotating, and distorting of the tube during compression. As the load is released, the compressed carbon tube springs right back to its original shape, showing elasticity and flexibility.

Fig. 3 Compressive mechanism of CTAs. (a) *in situ* TEM observation of carbon tubes during uniaxial loading and unloading. (b) *In situ* TEM observation of a single carbon tube during bending.

2. Cross linked point of carbon tube should be characterized in detail.

The TEM images of cross-linked joint of two carbon tubes are shown in Supplementary Fig. 6. It is observed that these two tubes have been “welded” together and there is an overlapped regime in the joint. We speculate that the drop height between two original nanowire templates results in this overlapped regime. As illustrated in Supplementary Fig. 6e, when carbon layers grew on the two nanowire templates and then encountered, only those atoms in the same plane connected with one another and formed bonds. The left parts of carbon layers continued to grow along the templates, forming a cross structure between tubes. The well interconnected structures in the CTAs might play a key role in their mechanical properties. We have added the above interpretation into the manuscript.

Supplementary Fig. 6. Microstructure of the joint between two carbon tubes. (a) TEM image of the joint. (b) EDS mapping of the joint. (c) High-resolution TEM image of the joint where an overlapped regime in the cross structure is seen. (d) EELS spectrum curves of a single tube with energy loss ranging from 50–150 eV and 1850–2000 eV, which suggest no Si element in the tubes. (e) Illustration of forming overlapped regime in the joint of carbon tubes.

3. More evidence for the sp^2 fraction and sp^3 fraction should be provided. EELS mapping of carbon tubes in this work could not give a strong support for the sp^2 fraction. Thanks for your comment. We agree with you that EELS mapping cannot give a precise and strong support for the sp^2 fraction. In fact to date, it is still very difficult to precisely

calculate the fraction between different bonds in a carbon material (that is why we carried out EELS mapping since the other method such as calculating the ratio of sp^2/sp^3 integral areas in XPS is much more inaccurate). Anyway, EELS mapping can at least give some direct and useful information about π - and σ -bonds for our material. We have changed our statements for this part and now it is: It is noteworthy that, these sp^2 and sp^3 of carbon systems correspond to very defined orbital states.³² Sp^2 is the hybridization of the atomic $2s$ orbital with $2p_x$ and $2p_y$ orbitals, resulting in a system of three planar σ -bonds forming an angle of 120° . The remaining $2p_z$ orbital that is perpendicular to the plane of the σ -bonds, forms a π -bond. Sp^3 corresponds to the hybridization of the atomic $2s$ orbital with its three $2p_x$, $2p_y$ and $2p_z$ orbitals in equal proportion and results in a system of four σ -bonds at an angle of 109.5° .³³ Related to these bonding states are the unoccupied anti-bonding states σ^* and π^* . As can be seen in Fig. 1h, electron energy loss spectroscopy (EELS) mapping shows that the π^* fraction varies between 0.32–0.45 and the distribution of π^* and σ^* bands along the tube are roughly homogenous (selected typical raw curves of C-K spectra are given in Supplementary Fig. 5). That might suggest a uniform mix of sp^2 and sp^3 bonds in the walls of carbon tubes.

Supplementary Fig. 5. Typical EELS spectra obtained from a single carbon tube. (a) EELS mapping with pixels that shows the ratio of π^* to σ^* using “two-window intensity-ratio” method. (b) Raw curve data drew from the selected pixels, which show different peak intensity of π^* and σ^* .

4. It is no big meaning to compare the ultimate stress versus maximum strain of CTAs compared to other elastic materials. They should compare the stress versus at a different certain strain of CTAs corresponding to other elastic materials.

Thanks for your comment. We have revised Fig. 2g. Now it shows the certain strain vs. corresponding stress retention for all materials.

Fig. 2 Mechanical properties of CTAs. (g) Compression curves of CTAs ($\sim 12 \text{ mg cm}^{-3}$) with strain ranging from 10–99%. Inset: Enlarging the onset of compression curve, indicating no permanent deformation. (b) Experimental snapshots of CTAs under uniaxial compression to 99% strain. (c) Near-zero ν of CTAs at different compressive strain (0–90%). Inset shows corresponding *in situ* SEM observation. (d) Compressive stress–strain curves of CTAs at 99% strain for 1,000 cycles. (e) Residual strain, stress, and energy loss coefficient of CTAs for different compressive cycles. (f) Compression cycle versus strain of CTAs compared to other elastic materials such as graphene aerogels,^{7,9,11,17,34,38,39} fiber aerogels,^{8,12,35-37,40} wood-derived sponges,⁴¹⁻⁴³ carbon tube aerogels,^{5,16} GO foams,^{10,44-46} and ceramic aerogels.^{18,47-52} (g) Compressive strain versus stress retention of CTAs during cycles compared to previously reported materials. The cycle numbers are marked next to the symbols.^{5,7-13,16-18,34-52} Detailed information can be seen in Supplementary Table 1.

Reviewer comments, second round –

Reviewer #1 (Remarks to the Author):

The authors have satisfactorily addressed all of my comments in the revised manuscript. I recommend this paper for publication.

Reviewer #2 (Remarks to the Author):

The revised manuscript looks better now. However, the template based CVD approach has been extensively reported for many years. The current work is really costly and no much enhanced performance is available. So, after reconsidering the key ideas and performance, I still reserve the opinion that I do not recommend the acceptance.

REVIEWER COMMENTS

Title: Carbon tube aerogels with high cross-links for super-elasticity and fatigue resistance

Manuscript ID: NCOMMS-23-01579B

Reviewer #1 (Remarks to the Author):

The authors have satisfactorily addressed all of my comments in the revised manuscript. I recommend this paper for publication.

Reviewer #2 (Remarks to the Author):

The revised manuscript looks better now. However, the template based CVD approach has been extensively reported for many years. The current work is really costly and no much enhanced performance is available. So, after reconsidering the key ideas and performance, I still reserve the opinion that I do not recommend the acceptance.

Point-by-Point Response to the Reviewers' Comments

Reviewer #1 (Remarks to the Author):

The authors have satisfactorily addressed all of my comments in the revised manuscript. I recommend this paper for publication.

Thank you very much.

Reviewer #2 (Remarks to the Author):

The revised manuscript looks better now. However, the template based CVD approach has been extensively reported for many years. The current work is really costly and no much enhanced performance is available. So, after reconsidering the key ideas and performance, I still reserve the opinion that I do not recommend the acceptance.

Thanks for your comment. First of all, as we mentioned in the main text, cross-links play a crucial role in the mechanical properties of our materials. However, other reported methods, have their limitations in preparing such highly cross-linked aerogels. For instance, a commonly used technique, freeze-drying that utilizes graphene or carbon nanotube powders as raw materials, often results in aggregation issues, particularly in high-concentration solutions. Also, the carbon constituents have a tendency to stack rather than forming proper connections and bonds.

To overcome this problem, we applied the sacrificial template and CVD to prepare high-performance our CTAs. Indeed, there is literature using this method (such as one of our referred paper *Science*. 363, 723–727 (2019) in which graphene is used as sacrificial template to fabricate BN aerogels). What makes our work different from the others is that we utilized high-density, hot-pressed nanowire aerogels ($200\text{--}300\text{ mg cm}^{-3}$) as raw template. These nanowire aerogels have sufficient cross-links yet exhibit brittle under large strain as seen below. Interestingly, when we deposited carbon layers and then removed the nanowire cores, good elasticity is achieved for our CTAs (see figure 2a, d, f, g). For deeper understanding the mechanism, we carried out in situ TEM observations and MD simulations, demonstrating how the hollow tubes deform and what role the sp^3 bonds play during deformation. We think our study is useful for the research community of carbon materials.

As for the expense, we acknowledge that it is more costly than direct stripping of graphite, as the ceramic nanowires have been all removed out and only carbon is left. But as we stated above, CVD has its unique advantage over other methods which use graphene powders as raw materials. In addition, we have realized scalable fabrication of ceramic

nanowire aerogels in 2019 (please see ref. R1). We are now also promoting the industrialization of these ceramic nanowire aerogels; therefore, the preparation cost of our template is not too high to be acceptable.

Fig. S1. Mechanical properties of the raw hot-pressed nanowire aerogels

Reference

1. Lu, D. et al. Scalable fabrication of resilient SiC nanowires aerogels with exceptional high-temperature stability. *ACS Appl. Mater. Interfaces*. **11**, 45338–45344 (2019).